# MASKS CAN BE DISTRACTING: ON CONTEXT COMPREHENSION IN DIFFUSION LANGUAGE MODELS

## ABSTRACT

Masked Diffusion Language Models (MDLMs) have emerged as an alternative to autoregressive language models, with a denoising objective that in principle enables more uniform context utilisation. We study the context comprehension of MDLMs and identify two key limitations. First, despite a more global training objective, MDLMs exhibit a **strong locality bias**: performance depends heavily on the proximity of relevant information to the prediction target. Second, we show that appending **mask tokens—required for generation—can substantially degrade context comprehension**. Through systematic ablations, we find that these masks act as distractors, impairing the model's ability to process relevant context. To mitigate this effect, we propose a **mask-agnostic loss** that enforces prediction invariance to the number of appended masks. Fine-tuning with this objective significantly improves robustness. Overall, our results reveal important shortcomings of current MDLMs and suggest concrete directions for improving context comprehension.

## 1 INTRODUCTION

Diffusion Language Models (DLMs) have emerged as a promising alternative to autoregressive language models (ARLMs), enabling parallel generation and bidirectional context modelling via iterative denoising (Austin et al., 2021; Lou et al., 2023). Among these, masked DLMs (MDLMs) (Sahoo et al., 2024; Shi et al., 2024) have scaled rapidly, achieving competitive performance and inference speed on standard benchmarks (Nie et al., 2025; Song et al., 2025). However, it remains unclear how MDLMs use context at inference time, and whether their distinct training objective mitigates the well-known inductive biases of ARLMs (Liu et al., 2023; Barbero et al., 2024). In this work, we present a systematic study of context comprehension in MDLMs and identify limitations with direct implications for training, evaluation, and deployment.

We demonstrate that, despite their global denoising objective, MDLMs do not use context uniformly. Instead, they exhibit a strong locality bias, relying disproportionately on information near the prediction target. Moreover, we find that generation-time design choices—particularly the number and placement of mask tokens—can substantially affect performance. This sensitivity stems from a core design feature of MDLMs: mask tokens are used both during training, via randomised masking, and at inference, to delimit the prediction span. Although intended as neutral scaffolding, we show that masks can act as distractors, impairing context processing. In MDLMs trained from scratch, this results in an inverse scaling effect: appending more mask tokens to the input consistently degrades performance. This degradation is not merely quantitative but also alters locality biases of the models, highlighting the critical and underappreciated role of masks in shaping MDLM behaviour.

In this work, using systematic empirical analysis, we make the following contributions towards identifying and explaining the fundamental limitations of MDLMs:

- **Locality bias (Section 3)**: We provide the first systematic evidence that MDLMs exhibit a strong *locality bias*, prioritising information near the masked token.
- **Inverse scaling with masks (Section 4)**: We uncover an *inverse scaling law with extra masks*: in MDLMs trained from scratch using the masked diffusion objective, additional mask tokens can significantly degrade performance, especially in long-context settings.
- **Mask-agnostic fine-tuning (Section 5)**: We propose a *mask-agnostic objective* that enforces prediction invariance to mask count, improving robustness of MDLMs.

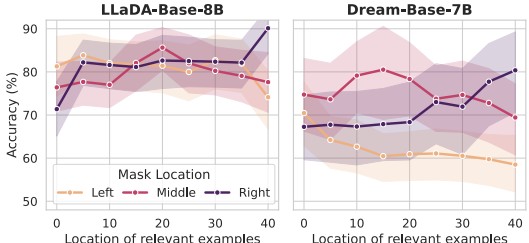

Figure 1: **MDLMs display a recency bias.** The performance of both MDLMs (LLaDA and Dream) and ARLMs is sensitive to the placement of relevant information within the context.

## 2 EXPERIMENTAL SETUP

We focus on open-source MDLMs to retain full control over generation settings. Specifically, we compare **LLaDA-8B** with Llama-3-8B (AI@Meta, 2024) and **Dream-7B** with Qwen-2.5-7B (Yang et al., 2024; Team, 2024), the ARLM used to initialise Dream. While LLaDA is trained from scratch using the masked diffusion loss (Sahoo et al., 2024), Dream represents an interpolation between ARLMs and MDLMs through AR initialisation. We prioritise accuracy over diversity and therefore use greedy decoding throughout. Additional results on **LLaDA-MoE** (Zhu et al., 2025) as well as LLaDA- and Dream-Instruct, are reported in Appendix C.1.

To evaluate context comprehension within the context limits of these models (LLaDA: 4096 tokens; Dream: 2048 tokens), we design a suite of few-shot multiple-choice tasks inspired by Todd et al. (2023), where models must infer abstract rules from examples. We construct 8 relevant word-based tasks (e.g., choose adjective, choose colour) and 2 number-based distractor tasks, enabling controlled manipulation of the placement of relevant information. Combining relevant and distractor tasks yields 16 evaluation tasks, each with 1000 test points. We report additional results on HotPotQA and a multidimensional classification dataset in Appendix C, with full experimental details in Appendix E.

## 3 ARE MDLMS LOCATION-SENSITIVE?

**Motivation.** ARLMs are known to exhibit locality biases (e.g., recency bias) limiting effective use of long-range contexts (Sun et al., 2021; Liu et al., 2023; Kossen et al., 2023). These effects are commonly attributed to the autoregressive loss, which prioritises recent tokens due to its sequential nature (An et al., 2024; Barbero et al., 2024; Bachmann & Nagarajan, 2024). MDLMs provide a natural testbed for assessing whether such biases are intrinsic to language modelling or arise specifically from the AR objective: they denoise tokens across the entire sequence in parallel and have been shown to be equivalent to any-order autoregressive models (Shuchen et al., 2025). We therefore investigate whether the diffusion objective alleviates locality biases, or whether they persist despite the change in training loss.

**Is the performance of MDLMs sensitive to the location of relevant information?** To assess whether MDLMs use context uniformly, we vary the position of a fixed block of relevant examples within a prompt containing additional distractors and measure accuracy on a held-out test question, placed on the right end of the context. As shown in **??**, MDLMs exhibit a strong sensitivity to information placement: performance is highest when relevant examples appear immediately before the test question and

Figure 2: **MDLMs prioritise information placed closest to the mask.**

degrades monotonically as they move farther away, indicating a *recency bias*. Gradient attribution analysis in Appendix C.2 provides complementary mechanistic evidence for this bias.

**Is the observed recency bias driven by absolute position or by proximity to the mask?** To disentangle these factors, we vary the position of the masked test question within the prompt while keeping the surrounding context fixed. As shown in Figure 10, performance is consistently highest when relevant information is located near the masked token, regardless of its absolute position in the input. This demonstrates that the identified recency bias reflects a more general **locality bias**: MDLMs prioritise information close to the prediction target rather than near the right edge of the context. We hypothesise that this locality bias arises from the masked diffusion objective itself—which disproportionately emphasises training instances with only a small number of masked tokens, where predictions can often be resolved using nearby context (Sahoo et al., 2024; Sharan et al., 2016).

## 4 THE DISTRACTING EFFECT OF EXTRA MASKS

**Motivation.** The previous section established that MDLMs do not process context uniformly; instead, they prioritise information closest to the mask. However, our analysis so far has been restricted to single-token answers, for which we allocated *a single mask token* during decoding. We now study how context comprehension changes when additional mask tokens are introduced—a question intrinsic to MDLM generation and largely unexplored in prior work.

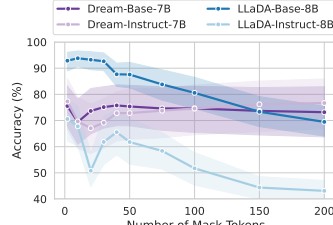

**Performance Degrades with Extra Mask Tokens.** We append varying numbers of masks to prompts (reflecting scenarios where the correct answer length is not known) and evaluate accuracy on the first (target) mask under single-step decoding. As shown in fig. 3, for LLaDA models (trained from scratch) *performance consistently degrades as more masks are added*. This means that beyond increased uncertainty, the masks induce a systematic shift of the model's predictions toward incorrect answers. Dream models (initialised from an ARLM) are more robust but still exhibit a measurable drop in accuracy, indicating partial but incomplete invariance. Thus, extra mask tokens, meant to act just as scaffolding for generation, can actively impair MDLM performance.

Figure 3: **Performance of LLaDA decreases significantly with added mask tokens,** while Dream is more robust.

**Are extra masks more harmful when long-range context is required?** To test whether mask-induced degradation is linked to impaired context comprehension, we vary the number of distractor examples– thereby increasing effective context length–while keeping the relevant information fixed. As shown in Figure 11, for LLaDA the performance gap grows as more distractors are added, indicating that masks disproportionately harm tasks requiring longer-range context integration. Additional evidence in Appendix C.3 shows that tasks requiring more context are more vulnerable to mask-induced degradation.

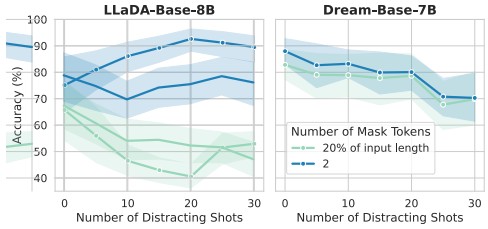

Figure 4: **For LLaDA, performance degrades more as the context length increases.**

**Is the Degradation Caused by Repeated Tokens?** To test whether mask-induced degradation is specific to mask tokens rather than a consequence of appending many identical tokens (which might be out of distribution for the model), we repeat the extra-mask experiment but replace masks with repetitions of a neutral token sequence (" ."). As shown in **??**, this manipulation has only a minor effect on LLaDA performance compared to the substantial degradation caused by extra masks. Hence, the observed performance decline is driven specifically by mask tokens, rather than by token repetition alone. For Dream, the effects of masks and dots are more similar, consistent with its greater robustness to mask-based perturbations.

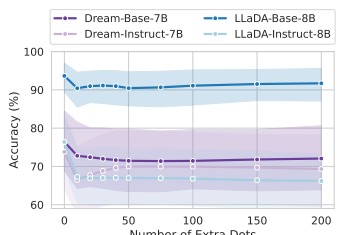

Figure 5: **Extra dots do not degrade performance as strongly as extra masks.**

**Can the Negative Effect be Fixed by Unmasking?** To test whether mask-induced degradation can be mitigated at inference time, we apply iterative unmasking–consistent with the MDLM denoising paradigm. As shown in Figure 12, unmasking substantially recovers the accuracy lost due to extra masks, with high-confidence unmasking strategy consistently outperforming random selection, particularly as the number of masks increases. This result supports the interpretation that extra masks act as distractors: progressively removing them (even with imperfect generations) restores the model's ability to focus on relevant context. However, unmasking incurs additional latency due to repeated decoding passes, limiting its practicality in low-latency or hardware-constrained settings.

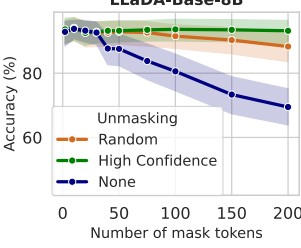

Figure 6: **Unmasking recovers accuracy lost to mask-induced distraction**.

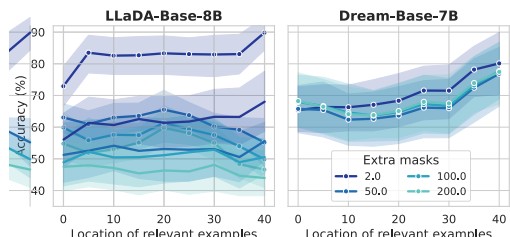

Figure 7: **Extra masks diminish the locality bias.**

**Do Extra Masks Affect Locality Bias?** To assess whether additional masks alter the locality bias observed in MDLMs, we repeat the information-placement experiment while appending varying numbers of extra mask tokens. As shown in Figure 13, extra masks degrade performance across all positions –although accuracy becomes more uniform across positions, this uniformity reflects consistently poorer performance rather than improved global context integration.

# 5 REDUCING THE DISTRACTING EFFECT THROUGH MASK-AGNOSTIC SFT

**Motivation.** In practice, the correct answer length is often unknown, making robustness to the number of mask tokens a desirable property for MDLMs. We therefore propose a supervised fine-tuning scheme that enforces invariance to the number of appended masks. This approach also provides mechanistic validation of our hypothesis that extra masks act as distractors: teaching the model to ignore them restores performance.

**Mask-Agnostic Loss.** To promote invariance to the number of appended mask tokens, we introduce a mask-agnostic (MA) loss. Given a prompt–answer pair, we construct two inputs that share the same prompt and partially masked answer but differ only in the number of additional mask tokens appended at the end. The MA loss combines two terms computed over masked answer positions: (i) a cross-entropy loss that enforces correct prediction of answer tokens under both masking configurations, and (ii) a TV-distance loss that explicitly penalises differences between the model's predictive distributions across the two inputs. The first term ensures accuracy regardless of mask count, while the second explicitly encourages prediction invariance to appended masks. The final objective is a weighted sum of these components, and full mathematical details and pseudocode are provided in Appendix D.

Figure 8: **MA loss rectifies the effect of extra masks.**

**Training details.** We fine-tune LLaDA models using LoRA adapters on a subset of the OpenOrca dataset, which differs from our in-context learning evaluation tasks and thus discourages overfitting to task-specific structure. We compare the proposed mask-agnostic (MA) loss against an ablated variant using cross-entropy (CE) only, training for approximately 1.2k gradient steps. Full training details are provided in Appendix D.

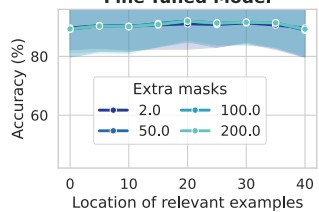

Figure 9: **MA loss reduces the locality bias** of LLaDA-Base.

**Results.** As shown in **??**, MA fine-tuning substantially improves robustness to variations in the number of appended mask tokens for both LLaDA-Base and LLaDA-Instruct, an effect not achieved by CE alone and also observed in LLaDA-MoE (Appendix C.1). The MA loss reduces prediction entropy and smooths model outputs as a function of mask count (Figure 31), yielding robustness comparable to iterative unmasking but with only a few decoding steps (Appendix C.4). In addition, MA fine-tuning reduces the locality bias of LLaDA models (Figure 9), indicating that sensitivity to extra masks is a correctable training artifact rather than a fundamental architectural limitation; corresponding results for LLaDA-Instruct are reported in Appendix C.5.

> **Takeaways.** Our results reveal a practical limitation of MDLMs: while parallel generation requires initialising inputs with many mask tokens, these masks themselves can act as distractors and impair context comprehension, independently of the degradation caused by fewer decoding steps. We refer to this additional cost as the "mask tax", and argue that it should be *explicitly considered* when designing robust fast samplers. The mask tax also has implications for evaluation: benchmark reports should clearly state the number of mask tokens used, and mask-sensitivity analysis–particularly on long-context tasks–should be incorporated as a standard component of MDLM evaluation, to assess robustness under realistic generation settings.

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

## A  RELATED WORKS

**Diffusion Language Models.** Diffusion Language Models (DLMs) have recently emerged as a promising alternative to the dominant autoregressive paradigm for text generation (Nie et al., 2025; HKU NLP Group; Song et al., 2025; Khanna et al., 2025). Unlike GPT-style models that generate text sequentially, DLMs employ an iterative denoising process that starts from a noisy representation and progressively reconstructs coherent text, enabling parallel token generation and bidirectional context modelling. While early research explored both continuous and discrete diffusion formulations for text, the discrete masked diffusion objectives (Sahoo et al., 2024; Lou et al., 2023; Austin et al., 2021; Shi et al., 2024) have recently dominated the landscape, allowing DLMs to effectively scale to larger model sizes and achieve competitive perplexity on standard benchmarks (HKU NLP Group; Nie et al., 2025). MDLMs have attracted a lot of attention for their potential to speed up inference (Kim et al., 2025; Frans et al., 2025; Song et al., 2025; Israel et al., 2025; Wu et al., 2025; Park et al., 2024; Agrawal et al., 2025) and improve controllability (Rector-Brooks et al., 2024; Gaintseva et al., 2025; Pani et al., 2025). However, to the best of our knowledge, a comprehensive evaluation of the influence of the masked diffusion training objective on the models' context comprehension abilities is still missing.

**Context Comprehension in Language Models.** Language models do not process information provided in the input uniformly (Sun et al., 2021; Qin et al., 2022; Barbero et al., 2024). Two well-documented position biases are primacy bias—a tendency to favour information appearing early in the input—and recency bias, where information near the end is weighted more heavily. These effects combine to produce the characteristic U-shaped accuracy curve in autoregressive models, often referred to as the *lost-in-the-middle* phenomenon (Liu et al., 2023). Empirical evidence for these biases comes from variations of the needle-in-the-haystack experiments (Kamradt) and related benchmarks across diverse tasks, including information retrieval (An et al., 2024), multi-document question answering (Liu et al., 2023), graph reasoning (Firooz et al., 2024), and in-context learning (Kossen et al., 2023). Primacy bias has been often attributed to the causal attention mask (Barbero et al., 2024), while recency bias has been linked to the training data distributions and the next-token prediction objective (Sharan et al., 2016; Barbero et al., 2024; An et al., 2024).

Whether MDLMs–trained on similar text corpora but with a fundamentally different objective–exhibit comparable position biases remains an open question. Prior work has explored related but distinct aspects of MDLMs: Liu et al. (2025) evaluated LLaDA on needle-in-the-haystack tasks to study *generalization to unseen context lengths*, but their setup was too simple to reveal recency effects within the models' context lengths. Similarly, Shansan et al. (2025) examined the "AR-ness" of MDLMs, defined as a *preference for left-to-right decoding*. In contrast, our work investigates whether MDLMs display AR-like tendencies in *processing* the context, rather than in their decoding strategy.

## B  EXPERIMENTAL DETAILS

**Setup for Experiment in Figure 1.** To assess whether model performance depends on the position of relevant information, we systematically vary the location of the *relevant* in-context learning examples within the prompt and measure the resulting accuracy on test questions. Specifically, we use 10 relevant examples (grouped together into one block), and 40 distractor examples. We keep the order of examples within the relevant and distractor groups fixed across all conditions, varying only the position of the relevant block within the overall sequence. We put the test example at the right end of the provided context, in an auto-regressive fashion.

**Extended Discussion of Results in Figure 1.** Figure 1 summarises the effect of information placement on model accuracy. Despite being trained with a masked diffusion objective–which does not enforce a strictly sequential prediction order–both MDLMs exhibit strong sensitivity to the position of relevant examples. Performance is highest when relevant information appears immediately before the test question, indicating a significant **recency bias**. Unlike ARLMs, which often display a U-shaped pattern (high accuracy when relevant examples are at the beginning *or* end of the prompt) (Liu et al., 2023; Barbero et al., 2024), MDLMs show a monotonic decline in accuracy as relevant information moves farther away. We do not observe a strong primacy effect in MDLMs, which aligns with the expectations, as the primacy effect has been attributed primarily to the causal attention mechanism (Barbero et al., 2024).

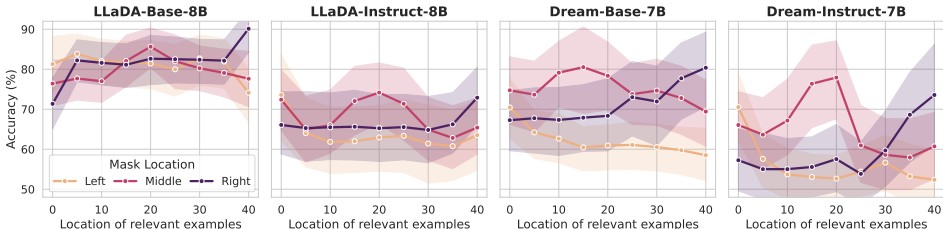

Figure 10: **MDLMs prioritise information placed closest to the mask.** All studied MDLMs perform best when relevant information is near the masked token, regardless of question position.

**Setup for Experiment in Figure 2.** The previous experiment revealed a strong recency bias in MDLMs, but it did not clarify its origin: does the bias arise because models generally prioritise information near the right edge of the context, or because they attend most strongly to the region around the mask token? To disentangle these factors, we repeat the previous experiment while varying the position of the test question (with its answer masked) within the prompt.

**Extended Discussion of Results in Figure 2.** Figure 2 shows that across all settings, model performance is highest when relevant information is placed *near* the masked question. This indicates that the previously observed recency bias is, in fact, a broader **locality bias**: MDLMs prioritise information close to the prediction target, regardless of its absolute position in the prompt. Interestingly, performance is consistently lowest when the masked question appears at the beginning of the input. We also note that for Dream, the performance is generally better when the relevant information is located *to the left* of the mask–suggesting a left-directed bias that resembles the behaviour of ARLMs that Dream was initialised from.

What is the source of the locality bias? Although MDLMs are trained on a more delocalised decoding objective than ARLMs, the masked diffusion loss is scaled by $1/p$, where $p$ is the probability of masking a token (Nie et al., 2025; Sahoo et al., 2024). Consequently, training places greater weight on cases where only few tokens are masked–scenarios where nearby context is usually sufficient for prediction, as in next-token prediction setting (Sharan et al., 2016). We hypothesise that this encourages MDLMs to rely on nearby context when processing the inputs.

**Setup of Experiment in Figure 3.** To measure how additional mask tokens affect MDLMs' context comprehension, we append varying numbers of mask tokens to the input prompt. We use 10 relevant and 40 distractor examples, mixing these two groups randomly together, to force the model to process the entire input context. In our format, the first mask token always corresponds to the answer for the test question. We decode the entire sequence in a single step but evaluate only the prediction for this first mask, ignoring all others. This setup isolates the effect of extra masks on the model's ability to correctly predict the target answer token, without introducing confounding factors from multi-step decoding.

**Extended Discussion of Results in Figure 3.** Contrary to our initial hypothesis—that additional masks might improve global reasoning—we observe a consistent performance degradation as the number of masks increases (fig. 3). This trend holds for both LLaDA-Base and LLaDA-Instruct. This is a surprising result: while extra masks could be expected to increase prediction entropy, inducing high uncertainty in generations, it is worrisome that for LLaDA models they lead to a consistent, monotonic degradation in accuracy even under greedy decoding—implying that with extra masks the mode of the model's token distribution is actively shifting to incorrect answers. For the base model, one plausible explanation is a distribution shift: during training, random noising rarely produces long unmasked prefixes followed by large contiguous mask spans. However, this scenario is not unusual for the instruct model, which exhibits a similar decline. Similar detrimental effect of too large numbers of masks was also seen in concurrent works (Li et al., 2025), although to a lesser extent.

Dream models appear more robust to large numbers of masks, but still exhibit a noticeable drop (6 and 8 percentage points for the base and instruct model, respectively) when approximately 20 masks are added, indicating that they are not fully invariant to the extra masks. We hypothesise that this difference may stem from Dream models being initialised from the weights of the autoregressive

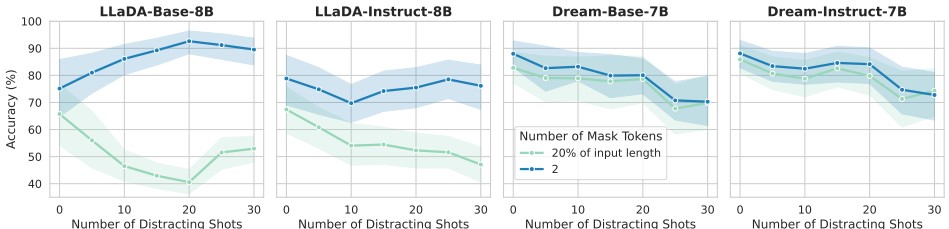

Figure 11: **For LLaDA, performance degradation becomes more significant as the context length increases.** We do not observe a similar effect for Dream, which is robust to the effect of extra masks.

Qwen-2.5, making mask tokens *less integral* to their architecture and training dynamics. In the following sections, we study this phenomenon of performance degradation due to extra masks in more detail, analysing several possible hypotheses which could explain this behaviour.

**Setup of Experiment in Figure 4.** We begin our analysis by investigating whether performance drop caused by extra masks is linked to impaired context comprehension. To that end, we examine how the effect of extra masks changes as the context length required to solve the task increases. Specifically, we vary the number of distractor examples in the prompt while keeping the number of relevant examples fixed, mixing the two groups together randomly. If extra masks indeed disrupt context processing, *we expect their negative impact to grow as more distractors are added*. This is because the model must filter relevant from irrelevant information over a longer context, and extra masks may disrupt its attention allocation.

**Extended Discussion of Results in Figure 4.** Figure 4 shows that for LLaDA, performance degradation due to extra masks generally increases with the number of distractors, and thus with the effective context length. This suggests that additional masks impair the model's context processing abilities.

We provide further evidence for this claim in Appendix C.3. There, we compare the degree of performance degradation caused by extra masks with the gains achieved when increasing the number of in-context examples across different tasks. We find a strong correlation: tasks that benefit most from additional context are also the most vulnerable to mask-induced degradation. This reinforces the conclusion that extra masks inhibit long-context comprehension.

**Setup for the Experiment in Figure 5.** In the previous section, we hypothesised that extra masks degrade performance because they act as distractors, drawing attention away from relevant context. To further validate this hypothesis and rule out alternative explanations, we test whether the performance degradation observed earlier is caused by the presence of the mask tokens specifically–rather than by simply appending many identical tokens. To evaluate this, we repeat the experiment from Figure 1 but replace the extra masks with a relatively neutral token sequence: the string `" . "` repeated multiple times. This ablation allows us to isolate the effect of mask tokens and verify that the observed behaviour is not merely due to an out-of-distribution repetition.

**Extended Discussion of Results in Figure 5.** Figure 5 shows that appending extra dots to the input has only a minor impact on the performance of LLaDA, especially compared to the substantial degradation seen in Figure 3 (for the dots, performance decreases by up to 3 and 10 percentage points for the base and instruct models respectively, compared to 23 and 27 percentage points for the masks). This confirms that in LLaDA the performance drop is driven by the presence of *masks* specifically, rather than by the mere repetition of identical tokens. For Dream, the effect of the masks and the dots are largely similar.

**Setup for the Experiment in Section Figure 6.** We examine whether the degradation caused by extra masks can be alleviated at inference time via iterative unmasking, that is, progressively resolving masked positions. This procedure is consistent with the denoising paradigm for which MDLMs are trained, where generation typically proceeds by unmasking the entire sequence. We run 40 decoding steps and compare two selection strategies for unmasking: choosing which tokens to unmask at random or according to the highest confidence.

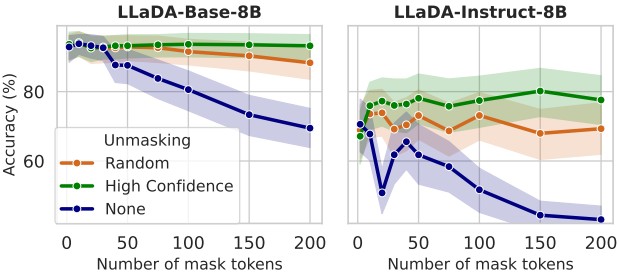

Figure 12: **Unmasking recovers accuracy lost to mask-induced distraction**. Unmasking strategies improve performance compared to no unmasking (None), with High Confidence consistently outperforming Random, especially as the number of masks increases.

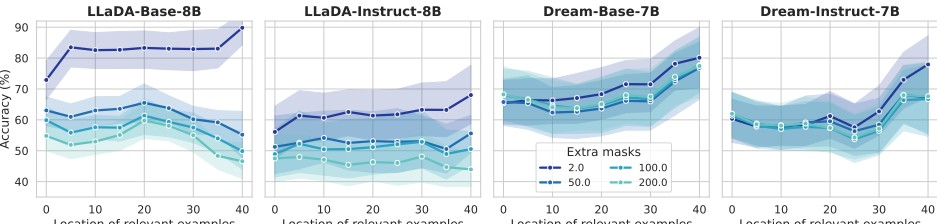

Figure 13: **Extra masks diminish the locality bias.** We measure performance sensitivity to the location of relevant information as extra masks are added. With more masks, accuracy becomes less location-dependent, mainly because it declines across all positions.

**Extended Discussion of Results in Figure 6 .** Figure 6 shows that unmasking (with 40 steps) markedly improves accuracy, recovering the performance lost due to the extra masks. This is especially true for the high-confidence unmasking strategy. This corroborates our findings in Figure 5: extra masks act as strong distractors and removing them, even with imperfect generations, restores focus on relevant context. While effective, this approach adds latency as it requires multiple decoding passes, which might not be desirable for specific hardware-constrained applications.

**Setup for the Experiment in Figure 7.** Finally, we revisit the question that motivated us to explore the effect of masks: can additional masks alter the locality bias observed in MDLMs (Figure 1)? To test this, we repeat the experiment from Figure 1 but append varying numbers of extra masks to the input. This allow us to assess how extra masks influence the model's ability to use information at different positions within the prompt.

**Extended Discussion of Results in Figure 7.** Figure 7 shows a surprising pattern: while extra masks degrade performance across all positions, the drop is more severe when relevant information is *closest* to the test question. As hypothesised, the performance becomes more uniform across positions, but this uniformity mainly reflects consistently poor results.

**Extended Discussion of Results in Figure 8.** Figure 8 shows that fine-tuning both LLaDA-Base and LLaDA-Instruct model with our MA loss allows to improve the performance of the models, making them more robust to variations in the number of masks appended to the input. Similar effects are visible in the LLaDA-MoE-Base (Appendix section C.1). The CE loss on its own does not have a similar effect, emphasising the importance of regularising generation with the TV loss directly. In Figure 31 we also show the effect of MA loss on the logits of the model, showing that our SFT procedure reduces the entropy of the model and makes it significantly smoother as a function of masks, thus increasing robustness. Unlike unmasking (Figure 6), which recovers accuracy through multiple decoding passes, our approach achieves similar robustness in only **a few decoding step** (we show performance improvements when using for 2, 4 and 6 decoding steps in Appendix C.4). This makes it attractive for low-latency applications and for distillation pipelines, where minimising generation steps is critical for efficiency and model compression.

Further, the success of our mask-agnostic loss offers additional insights into the nature of the sensitivity of LLaDA models to the extra masks: it proves that this behaviour is not an insurmountable architectural flaw, but rather a training artifact, which *can be corrected* by enforcing invariance to the number of mask tokens.

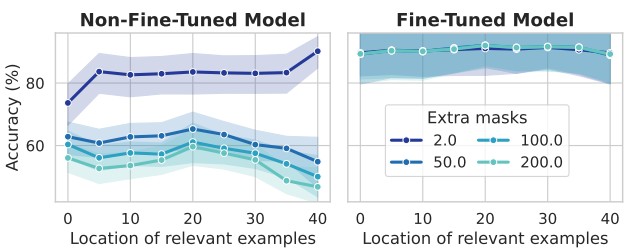

Figure 14: **MA loss reduces the locality bias** of LLaDA.

# C    ADDITIONAL EXPERIMENTAL RESULTS

## C.1    RESULTS ON LLaDA-MoE

**Motivation.**    As the area of MDLMs is still in early stages of development, the number of open-source MDLMs available for evaluation is still heavily limited. In our work, we follow the example of existing works and conduct all the evaluations on the LLaDA and Dream models (Shansan et al., 2025; Israel et al., 2025; Li et al., 2025; Wang et al., 2025). We believe that the identified limitations of these models, particularly given their prevalence, can guide training and deployment of future MDLMs and hence significantly contribute to the field. To improve the generalisability of our results, we have rerun the experiments in sections section 3 and section 4 of the paper also on LLaDA-MoE (Zhu et al., 2025)–a mixture of experts MDLM, providing significant training details in the provided model report. Importantly, **LLaDA-MoE has been fine-tuned to context lengths of 8k**, thus increasing the context length compared to LLaDA and Dream.

**Results.**    Figures 15-21 show the results of our analysis conducted on LLaDA-MoE. LLaDA-MoE largely displays patterns similar to that of LLaDA, although some results merit further discussion. In Figures 15 and 16 we note that LLaDA-MoE-Base does not display a significant recency bias (its performance is mostly agnostic to the location of relevant examples). We hypothesise that this might be because LLaDA-MoE was fine-tuned to handle context lengths of 8k, which is significantly more than the length of the tasks considered in our evaluation. Nevertheless, the gradient attribution analysis (Figure 24) still demonstrates patterns consistent with the recency bias present in other MDLMs, suggesting that this issue might still affect performance in tasks with longer input. **The fine-tuning experiment with the MA loss (fig. 21) clearly demonstrates that the MA loss can be effective in reducing the negative effect of extra masks**.

**Remark.**    We note that the performance of LLaDA-MoE presented in Figure 15 does not align exactly with the performance for the case when we use 2.0 masks in Figure 20. We note that this discrepancy stems from the fact that in Figure 15 we use only a single mask, followed by the end of sentence, rather than two separate masks (see Section E.4 for details). This discrepancy indicates that LLaDA-MoE is highly sensitive to the number of masks, and small variations can significantly affect performance, further reiterating the importance of our findings.

## C.2    GRADIENT ATTRIBUTION ANALYSIS OF MDLMs

### C.2.1    MEASURING THE LOCALITY BIAS IN MDLMs

**Setup.** To deepen our understanding of locality bias in MDLMs and ARLMs, we perform gradient attribution analysis (Lopardo et al., 2024), which quantifies how sensitive the model's prediction is to changes in each input token. Specifically, we compute the L2 norm of the gradients of the logit corresponding to the predicted answer token with respect to the input token embeddings. This provides a more mechanistic measure of each token's influence on the output. We use a dataset containing 10 relevant examples and 40 distractors, randomly mixed together to ensure the model must process the entire context to arrive at the answer. While the examples remain fixed across runs, their relative ordering is randomised across 30 seeds. If the models were location-invariant, gradient magnitudes would be roughly uniform across the positions. As the in-context examples do not change

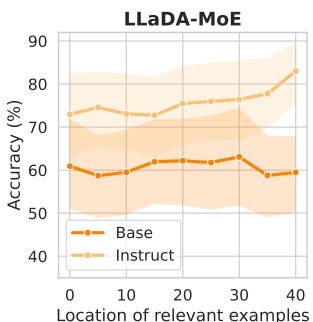

Figure 15: **Recency bias in LLaDA-MoE (re: Fig 1).** LLaDA-Moe-Instruct displays a strong recency bias, as seen also in other MDLMs, while the performance of LLaDA-MoE-Base is more agnostic to the location of relevant examples within the context.

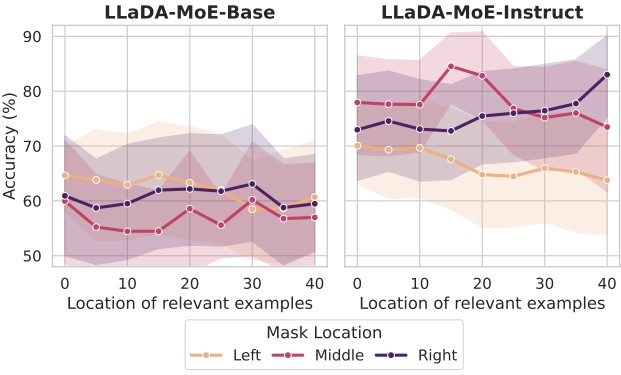

Figure 16: **Locality bias in LLaDA-MoE (re: Fig 2).** LLaDA-MoE-Instruct displays a strong locality bias, as seen also in other MDLMs (the performance is best when the relevant examples are located close to the masked question). The performance of LLaDA-MoE-Base is more uniform across the locations of relevant examples, although at a significantly lower level overall.

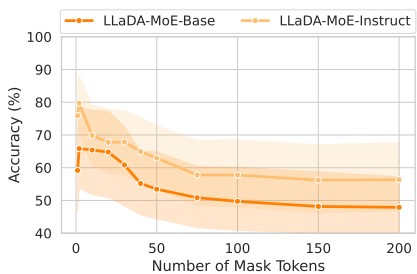

Figure 17: **Performance of LLaDA-MoE decreases significantly with added masks (re: Fig. 4).**

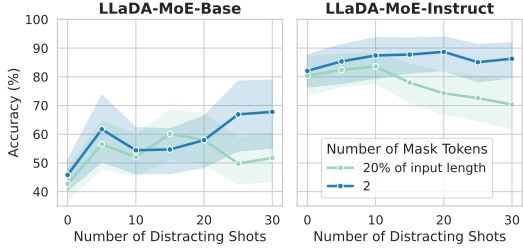

Figure 18: **For LLaDA-MoE, the performance degradation becomes more significant as the context length increases (re: Fig 5).** This effect is particularly visible in LLaDA-MoE-Instruct.

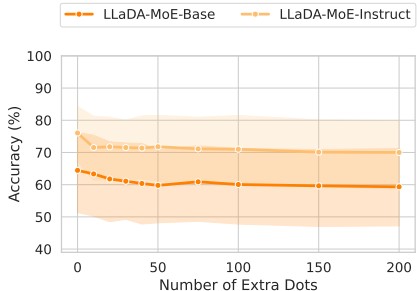

Figure 19: **For LLaDA-MoE, extra dots do not degrade performance as strongly as extra masks (re: Fig 6).**

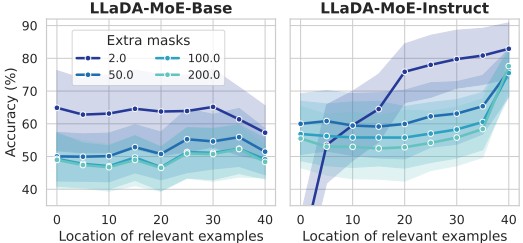

Figure 20: **Extra masks alter the locality bias in LLaDA-MoE (re: Fig. 8)**. For both the Base and the Instruct model, the performance becomes significantly worse as we add extra masks, across all locations. For LLaDA-MoE-Instruct in particular, the performance is more uniform across most locations with the extra masks.

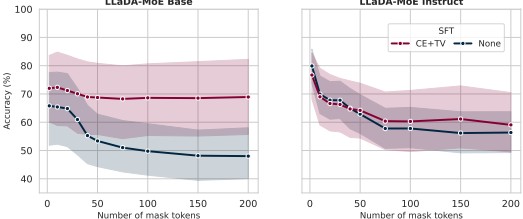

Figure 21: **MA loss largely rectifies the effect of extra masks.** Particularly in the LLaDA-MoE-Base model, fine-tuning with the MA loss allows to induce the robustness to extra masks, leading to improved performance. We use random unmasking strategy.

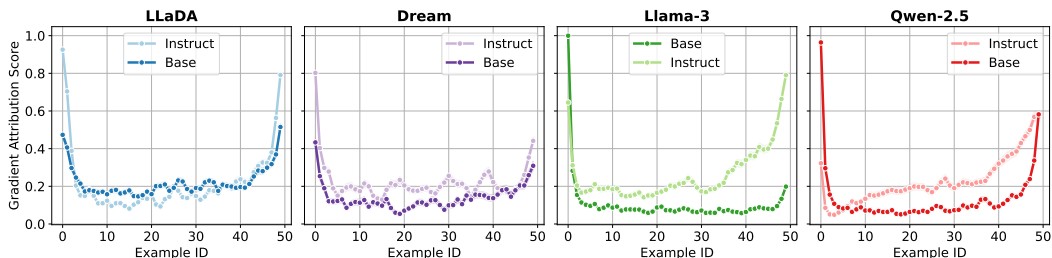

Figure 22: **Gradient attribution analysis further illuminates the locality bias of the models.** Although all models display the characteristic U-shaped behaviour, MDLMs demonstrate more uniform gradients across different positions, indicating reduced locality bias compared to their ARLM counterparts.

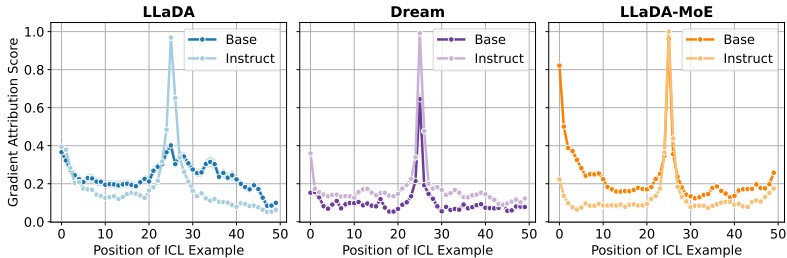

Figure 23: **Gradient attribution analysis confirms locality bias in MDLMs.** Normalised gradient attribution results for when the target question is placed in the **middle** of the input context.

for different test questions, for computational efficiency we evaluate the gradients for a sample of 20 test questions for each task only. Similarly to what we did in **??**), we consider three different locations for the masked target question: at the beginning, in the middle and at the end of the input context.

**Results.** Figure 22 shows *normalised* gradient scores across the different in-context examples. Consistent with earlier performance trends (**??**), all models exhibit a non-uniform pattern, forming the characteristic U-shape associated with primacy and recency effects. However, MDLMs display more uniform gradients than ARLMs, suggesting more global comprehension abilities. Notably, MDLMs also show less pronounced primacy bias compared to their autoregressive counterparts. Figures 25 and 23 further show a clear locality bias of the studied MDLMs: the normalised gradients have consistently larger values at positions closer to the mask of interest (i.e. for positions 20-30 when the masked question is located in the centre of the input, and for positions 0-10 when the masked question is located on the left end of the input). This provides additional evidence for our results presented in Section 3, indicating that MDLMs display a strong locality bias.

### C.2.2 GRADIENT ATTRIBUTED TOWARDS THE EXTRA MASKS

**Motivation.** To assess how strongly MDLMs prioritise the extra mask tokens over any other tokens in the input, we analyse gradient-based attributions. Using the configuration from **??**, we append 50 mask tokens to the input and measure the normalised gradient of the masked answer token (i.e., the first mask) with respect to all other tokens in the sequence. This quantifies the influence of the added masks on the model's prediction and the model's sensitivity to their presence relative to the surrounding context.

**Results.** Table 1 reports the average normalised gradients for three token groups: (i) the added mask tokens, (ii) the last 50 non-mask tokens closest to the target mask, and (ii) all non-mask tokens. Across all models, gradient magnitudes attributed to mask tokens are markedly higher than those attributed to either non-mask group. This pattern indicates that MDLMs allocate disproportionate attention to the added masks, consistent with our broader observation that these models are heavily influenced by the mask tokens at the expense of effective context utilisation. We also note that the

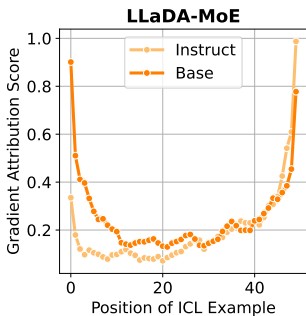

Figure 24: **Gradient attribution analysis reveals recency bias in LLaDA-MoE (re: Fig 3).** Both LLaDA-MoE-Base and LLaDA-MoE-Instruct display a strong recency and primacy bias, based on the gradient attribution analysis.

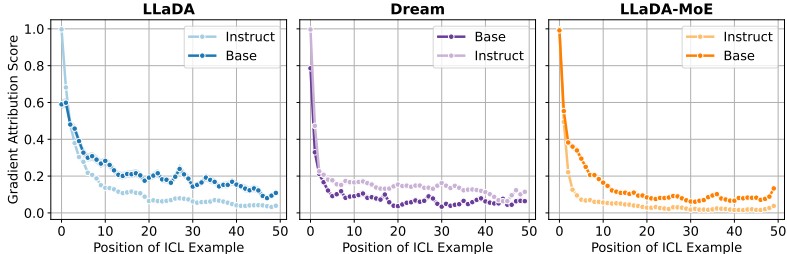

Figure 25: **Gradient attribution analysis confirms locality bias in MDLMs.** Normalised gradient attribution results for when the target question is placed on the **left** end of the input context.

| Model Name | Masks | Non-Masks (Last 50) | Non-Masks |
|---|---|---|---|
| Dream-Base-7b | $0.282 \pm 0.040$ | $0.012 \pm 0.007$ | $0.005 \pm 0.003$ |
| Dream-Instruct-7b | $0.144 \pm 0.031$ | $0.030 \pm 0.005$ | $0.018 \pm 0.002$ |
| LLaDA-Base-8b | $0.234 \pm 0.021$ | $0.005 \pm 0.002$ | $0.005 \pm 0.002$ |
| LLaDA-Instruct-8b | $0.220 \pm 0.031$ | $0.057 \pm 0.014$ | $0.017 \pm 0.003$ |
| LLaDA-MoE-Base | $0.237 \pm 0.034$ | $0.094 \pm 0.016$ | $0.029 \pm 0.003$ |
| LLaDA-Moe-Instruct | $0.188 \pm 0.032$ | $0.150 \pm 0.024$ | $0.028 \pm 0.004$ |

Table 1: **MDLMs are particularly sensitive to mask tokens.** We show the average normalised gradients attributed to the mask tokens, compared to all the other tokens in the input sequence.

last 50 non-mask tokens (i.e. the 50 tokens located directly to the left of the mask) have significantly higher gradient scores than non-mask tokens in general, reiterating the recency bias.

## C.3 CORRELATION BETWEEN MASK DEGRADATION AND CONTEXT SIGNIFICANCE

**Extra Masks Hurt Behaviour On Tasks Requiring Long Context Comprehension.** In **??**, we presented initial evidence that additional masks impair the model's ability to utilise long contexts. Here, We investigate this effect further by analysing the performance of MDLMs on a variety of few-shot learning tasks with single-token answers.

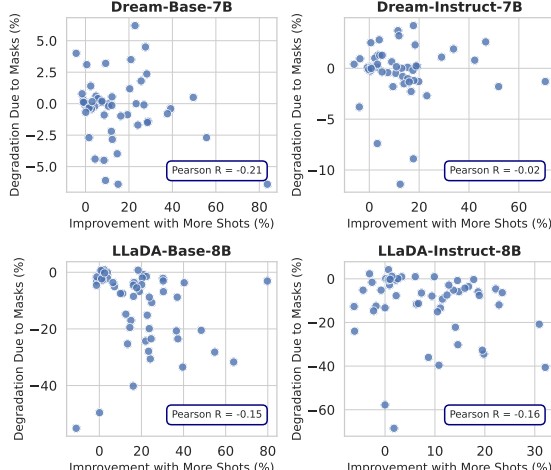

Figure 26: **For LLaDA models, tasks that benefit more from additional ICL shots exhibit stronger performance degradation under extra masks.** Dream shows no such trend, remaining more robust to extra masks.

**Setup.** For each task, we evaluate performance along two axes. First, we measure the gain in accuracy when increasing the number of in-context examples from 5 to 25, which serves as a proxy for the task's dependence on long-context information. Second, we compare performance between two masking configurations–one with a single extra mask and one with 200 extra masks–using the 25-shot setting. This quantifies the degradation in predictive accuracy induced by extra masks.

**Results.** Figure 26 visualises the relationship between performance gains from additional in-context examples and degradation due to extra masks (both expressed as absolute accuracy differences). For LLaDA-Base and LLaDA-Instruct, most points lie below the $y = 0$ line, indicating substantial degradation—up to 60% on some tasks. The negative Pearson correlations ($R = -0.15$ and $R = -0.16$, respectively) suggest that tasks benefitting most from longer contexts are also those most affected by extra masks. While the correlations are modest, they reinforce the hypothesis that masking disproportionately disrupts long-context processing, though other factors likely also determine the level of degradation.

By contrast, Dream models show minimal and less consistent degradation ($\leq 12\%$), aligning with our earlier observation that MDLMs initialised from autoregressive (AR) weights exhibit increased robustness to masking effects.

**Details of the few-shot learning tasks used.** Each point on the scatterplots presented in Figure 26 corresponds to a different few-shot learning task. We use the following few-shot learning datasets investigated in the different sections of the paper: (1) The pattern recognition tasks described in Section 2 (16 combinations). (2) All the variants of the multi-dimensional classification dataset described in Section C.8. Additionally, we use the following popular ICL datasets: AG News (Zhang et al., 2015), SST-2 (Socher et al., 2013), Rotten Tomatoes (Pang & Lee, 2005), as well as MRPC, RTE and QNLI from GLUE (Wang et al., 2018). For AG News, we restrict the dataset to three categories only (excluding 'Science and Technology') such that each of the correct labels can be expressed with a single token only. For RTE dataset, we use the original validation set for getting the in-context examples and use the train set as the evaluation set, to maximise the number of examples in the evaluation set. For datasets where the examples are ordered by label, we shuffle the datasets upon loading to ensure that there is an even distribution between the different classes within the in-context examples provided to the models. For RTE, QNLI and AG News datasets we also filter the examples in the train and test sets such that the length of the text does not exceed 500 characters.

## C.4 ROBUSTNESS ANALYSIS: DECODING WITH FEW STEPS ON THE LLADA MODELS

**Motivation.** Our results in Figure 12 demonstrated that when using 40 decoding steps, the performance degradation due to extra masks is alleviated. In Figure 14 we showcased that the same effect

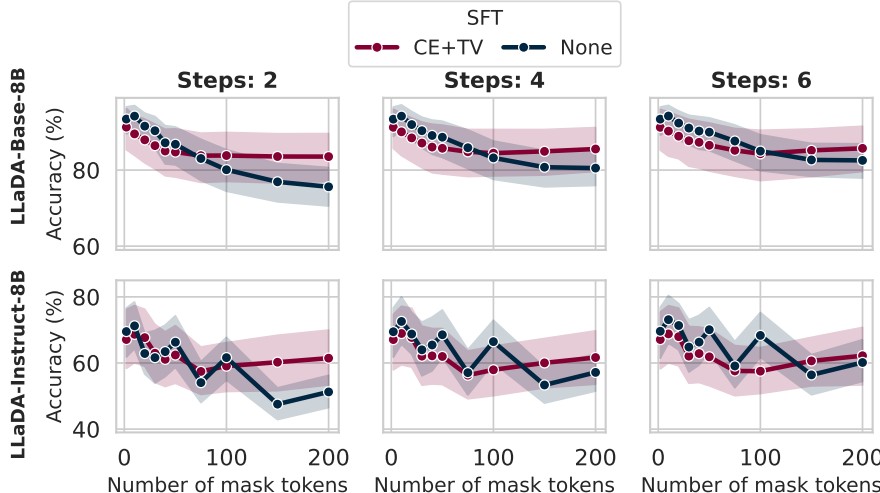

Figure 27: Performance across varying numbers of mask tokens for different decoding steps (2, 4, and 6) using random unmasking strategy, for LLaDA models. The base model (None) shows significant performance degradation as the number of mask tokens increases, while our CE+TV fine-tuned model maintains more stable performance across all configurations.

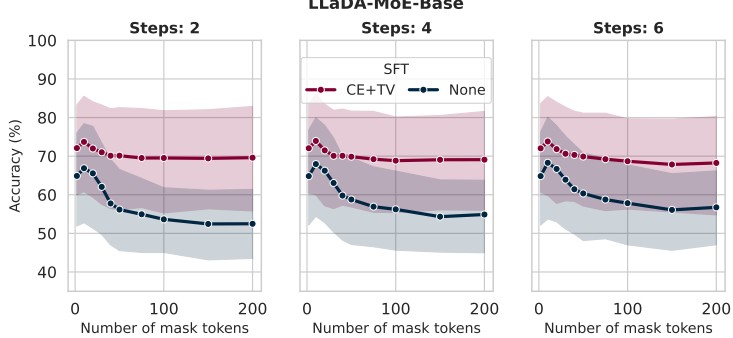

Figure 28: Performance across varying numbers of mask tokens for different decoding steps (2, 4, and 6) using random unmasking strategy, for LLaDA-MoE-Base model. The MA loss helps to rectify the negative effect of extra masks across all numbers of decoding steps considered.

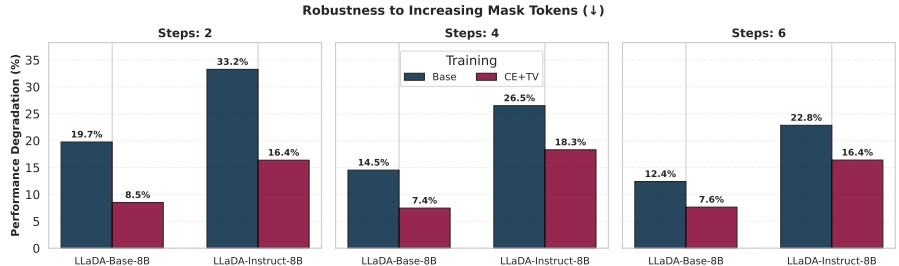

Figure 29: Relative performance degradation (measured as $\frac{\text{max accuracy} - \text{min accuracy}}{\text{max accuracy}} \times 100\%$) across different numbers of decoding steps. Lower values indicate better robustness to increasing mask tokens. Our CE+TV fine-tuning reduces degradation by 38-49% compared to the base model, demonstrating significantly improved robustness with minimal accuracy trade-offs.

can be achieved in just 1 decoding step with the help of our mask-agnostic fine-tuning, achieving much lower latency. Here, we provide intermediate results showing how performance varies when using 2, 4, and 6 decoding steps to further evaluate the benefits of using the mask-agnostic loss.

**Results.** Figure 27 shows the results obtained using the random unmasking strategy (tokens were unmasked in random order). Across all configurations, we observe that increasing the number of mask tokens leads to performance degradation in both the base model and our fine-tuned model. However, the CE+TV fine-tuned model consistently maintains higher performance and exhibits significantly less degradation.

**Robustness Analysis.** To quantify this improved robustness, we measure the relative performance degradation as the percentage drop from maximum to minimum accuracy: $\frac{\text{max accuracy} - \text{min accuracy}}{\text{max accuracy}} \times 100\%$. As shown in Figure 29, our CE+TV fine-tuning substantially reduces performance degradation across all step configurations:

- **LLaDA-Base-8B**: Degradation reduced from 15.5% to 7.9% (49% reduction)
- **LLaDA-Instruct-8B**: Degradation reduced from 27.5% to 17.0% (38% reduction)

Importantly, this improved robustness comes with minimal accuracy trade-offs. The CE+TV model maintains competitive or superior performance at low mask token counts while being significantly more robust as the number of mask tokens increases. This demonstrates that our mask-agnostic fine-tuning not only enables efficient single-step decoding but also fundamentally improves the model's ability to handle varying numbers of mask tokens, making it more practical for real-world applications where computational constraints may vary.

However, we emphasise that while our method mitigates the degradation due to extra masks, it does not fully eliminate it. The fact that a non-negligible performance drop persists–even after targeted fine-tuning and multiple decoding steps–underscores the severity of the mask distraction phenomenon. It suggests this is not a trivial artifact, but a deep-seated characteristic of current MDLM architectures that cannot be easily ignored and requires continued investigation.

## C.5 Additional Results for the Fine-Tuned LLaDA-Instruct

In Figure 30 we provide additional results visualising how the fine-tuning procedure affects the locality of the LLaDA-Instruct model, under different numbers of masks. We observe that the model is more robust to the extra masks, and its performance is more uniform over the different positions of relevant information.

## C.6 Confidence and Entropy as a Function of Masks

In Figure 31 we plot the effect of fine-tuning the LLaDA models with the MA loss on the confidence (calculated as the probability of the generated token, under the greedy decoding scheme) and the entropy of the model's generations. We observe that training with the MA loss significantly increases the confidence in the generated answer for the Base model, and makes the confidence more smooth as a function of extra masks for both models. Furthermore, MA loss also significantly decreases the entropy for both models, also making it more smooth.

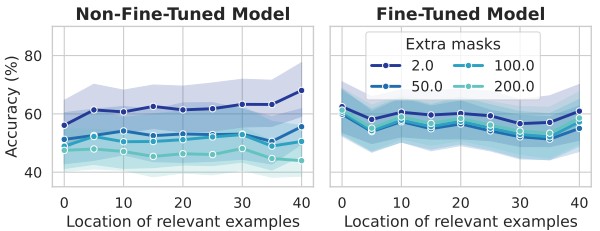

Figure 30: **MA loss (CE+TV) reduces the degrading effect of extra masks in LLaDA-Instruct, and removes the locality of model, however, at the cost of a slight performance decrease.**

## C.7 Experiments on the HotPotQA dataset

**Motivation.** To further apply whether our results generalise to other in-context learning tasks, beyond the few-shot learning setting, we use a subset of the HotPotQA dataset (Yang et al., 2018).

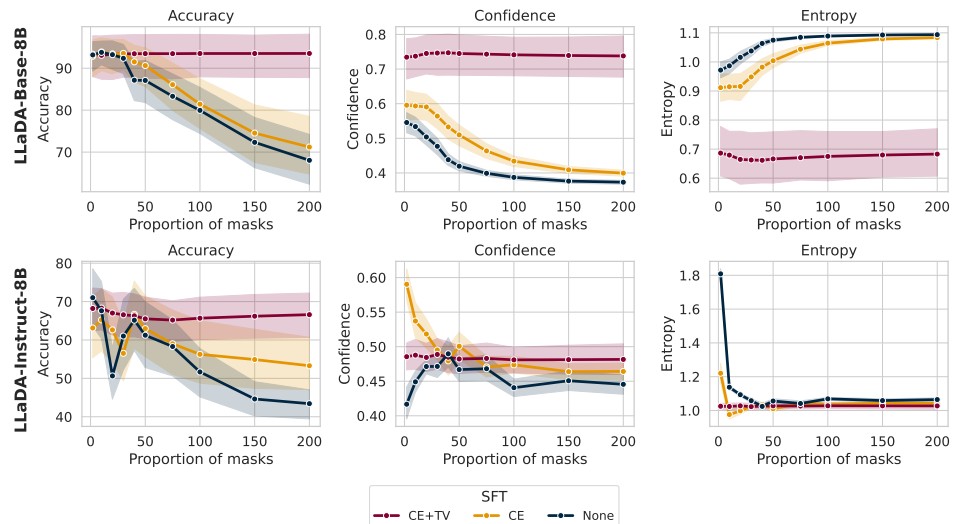

Figure 31: **MA loss (CE + TV) decreases the model's entropy and increases the confidence in the generated token**, while making both a smoother function of the number of extra masks, thus increasing the robustness of the model.

This dataset consists of Wikipedia-based question-answer pairs. The questions require finding and *simultaneously* reasoning over multiple supporting documents (facts), thus ensuring that the dataset requires long-context comprehension.

**Dataset.** We utilised the 'distractor' configuration of HotPotQA and loaded it via the Hugging Face datasets library.[1] Our preprocessing focused on extracting binary-choice questions by filtering for examples containing "or" in the question text. Using regular expression pattern matching, we parsed such questions to extract the question stem and two possible options (A and B). We applied additional filtering to remove examples with input lengths exceeding 1000 tokens (to fit within the context window of the studied MDLMs) and those that could not be reliably converted to multiple-choice format. This approach allowed us to work with a standardized set of binary-choice questions from HotPotQA with single-token answers that were suitable for our controlled experiments and could be reliably evaluated using the accuracy metric. For each example, we concatenated the provided supporting facts (context) together with the question:

```
f"**Context**:\n'{entry['context']}'.\n\n"
    + f"**Question**: '{entry['question']}'"
    + f" [A] {entry['option_A']}\n"
    + f" [B] {entry['option_B']}\n"
    + "**Answer**:[{entry['answer']}]"
```

We use a system prompt ("Which of the following answers is true? Respond with [A] or [B].") and append one in-context learning example to ensure that the model can correctly format its answer.

As the input lengths in this dataset are more variable, rather than adding a pre-determined number of masks as in previous experiments, we add a number of masks proportional to the number of tokens in the input text.

**Results.** We evaluated the performance of LLaDA-Base and LLaDA-Instruct, with and without the mask-agnostic fine-tuning. Without the fine-tuning, we observe a high sensitivity of the Base model to the number of extra masks, with performance decreasing sharply when the number of masks is equal to $\approx 5\%$ of the input length (which corresponds to 90-100 tokens). The fine-tuning allows to effectively remove the variability to the extra masks, once again smoothing out the confidence and entropy curves.

---

[1]https://huggingface.co/datasets/hotpotqa/hotpot_qa

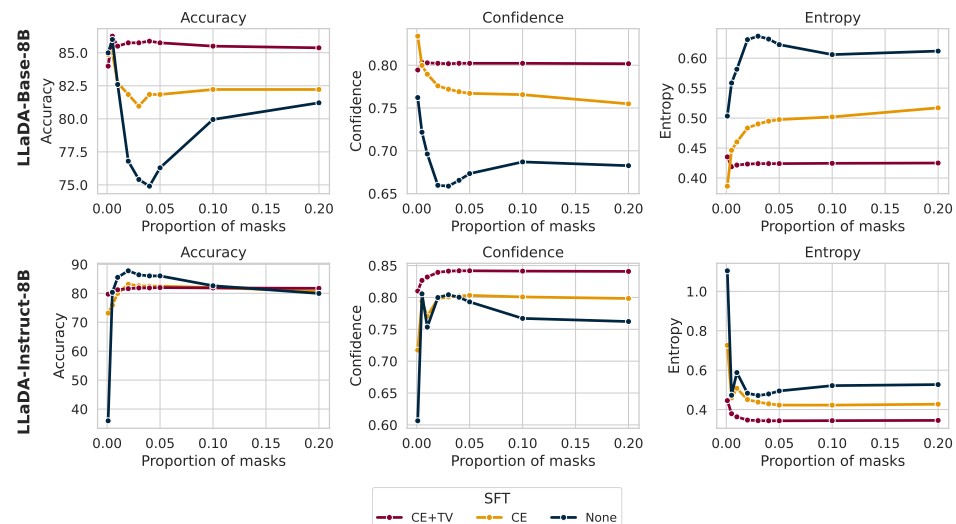

Figure 32: **On the HotPotQA dataset, the MA loss also improves the robustness of the models to the varying number of masks.** We observe improved performance particularly for the LLaDA-Base model.

For the Instruct model, we note that even before the fine-tuning the model is more robust to the number of extra masks in this setting. However, the MA loss still allows to smooth out the confidence and the entropy of the model. Further, the MA loss makes the model more robust in the case when the number of available tokens is small (1-2) tokens, in which case the original model fails to provide a coherent answer.

## C.8 EXPERIMENTS ON THE MULTI-DIMENSIONAL CLASSIFICATION DATASET

**Motivation.** While in the pattern recognition tasks presented in the main paper it is relatively clear which examples carry signal for the test question (number vs word tasks), we also consider the setting where this distinction is more blurry, and the contribution of each example to the answer is more ambiguous. Specifically, we construct a multidimensional classification task, where each point is described using a three-dimensional integer coordinates and a binary label. To make the tasks difficult, we use different non-linear decision boundaries, described below. The task of the model is to predict the label for a new point. To measure the sensitivity to the position of information, we manipulate the order in which the points are presented: ordering them either randomly, or by the L2 distance in the input space to the test point.

**Dataset.** To evaluate recency bias, we constructed several synthetic binary classification datasets with varying complexity. Each dataset was designed to present different learning challenges, from nonlinear decision boundaries to complex manifold structures. For reproducibility, we generated each dataset type with 5 different random seeds. We utilized four distinct dataset types in our experiments:

1. **Nonlinear dataset:** This dataset features nonlinear decision boundaries created through polynomial feature transformations. We first generated base features as random integers between 1 and 100. We then augmented these with squared terms and interaction terms between features, creating a nonlinear feature space. The final binary labels were determined by applying a logistic function to a weighted sum of these features (with randomly generated coefficients), followed by thresholding at 0.5.

2. **Swiss-roll dataset:** We employed scikit-learn's `make_swiss_roll` function to generate data points along a 3D swiss roll manifold. The continuous position along the roll (colour parameter) was converted to binary labels by thresholding at the median value, creating two interleaved classes that cannot be separated by a linear boundary. The 3D coordinates were then scaled to integers between 1 and 100 to maintain consistency with our other datasets.

3. **Moons dataset:** Using scikit-learn's `make_moons` function, we created two interleaving half-moon shapes in 2D space. This dataset presents a clear nonlinear boundary challenge. The resulting coordinates were scaled to integers between 1 and 100.

4. **Circles dataset:** We generated concentric circles using scikit-learn's `make_circles` function, creating another challenging nonlinear classification problem. As with the other datasets, the coordinates were scaled to integers between 1 and 100.

To ensure class balance, we generate equal numbers of positive and negative examples for both training (100 examples) and test splits (1000 examples) of each dataset. Additional dimensions beyond those generated by the base algorithms were filled with random integers, such that each dataset has is three-dimensional. Each input vector is stored as a space-separated string of integers. We use the class labels 'Above' and 'Below'.

**Setup.** To study the recency bias (i.e., whether or not the models have a tendency to pay more attention towards examples which are closer to the generation point), we employ the following ordering schemes to the selected in-context examples:

- **Random ordering:** The in-context examples are ordered randomly.
- **Ordered by distance to the test point, in decreasing order:** When formatting each prompt, we compute the L2 distance of each in-context example to the test example. We then order the in-context examples in decreasing order, such that examples on the far left of the prompt are furthest away from the test point, and examples on the far right are closest in distance to the test point. This corresponds to the 'Position of relevant information: Right' setting.
- **Ordered by distance to the test point, in decreasing order:** We again compute the L2 distance of each in-context example to the test example, but now order the points in increasing order, such that examples on the far left of the prompt are closest to the test point, and examples on the far right are furthest away in distance to the test point. This corresponds to the 'Position of relevant information: Left' setting.

We note that in all settings, the selected in-context examples are fixed, we just change their order within the prompt. Under this setting, a conventional supervised learning algorithm should be agnostic to the ordering of the provided information. We run the experiments with the masked example placed both on the left-end of the prompt and on the right-end of the prompt.

**Results.** Firstly, we use the created setup to further evaluate the locality bias of the MDLMs and ARLMs. Results in Figure 33 show that performance of MDLMs (particularly Dream, initialised with the weights of an ARLM) drops significantly when relevant examples are far from the masked question, confirming a locality bias – though weaker than in ARLMs.

Secondly, we evaluate the robustness of the LLaDA-Base and LLaDA-Instruct models to the varying numbers of masks under the different ordering schemes. We focus on the case with 30 in-context examples, with the test question located on the right-end of the in-context examples. Results in Figure 34 show that for the Base model, our MA loss prevents performance degradation, particularly for the random ordering and the ordering by increasing distance (where the most relevant information is far away from the test question). For the Instruct model, our fine-tuning scheme improves robustness to the number of masks, and consistently prevents significant performance degradation with small numbers of masks.

# D DETAILS OF THE SUPERVISED FINE-TUNING PIPELINE

## D.1 FORMULATION OF THE MASK-AGNOSTIC LOSS FUNCTION

To encourage invariance to the number of extra masks, we propose a **mask-agnostic (MA) loss**. Consider prompt–answer pairs, where $\boldsymbol{q} = (q^1, \ldots, q^{n_q})$ is the tokenised prompt and $\boldsymbol{a} = (a^1, \ldots, a^{n_a})$ is the tokenised answer. We construct a noised version of the answer, $\tilde{\boldsymbol{a}} = (\boldsymbol{1} - \boldsymbol{u}) \circ \boldsymbol{a} + \boldsymbol{u} \circ \boldsymbol{m}$, where $\boldsymbol{1} \in \mathbb{R}^{n_a}$ is the vector of 1s, $\boldsymbol{m} \in \mathbb{R}^{n_a}$ is the vector of mask tokens, and $\boldsymbol{u} = (u^1, \ldots, u^{n_a})$ is a vector of samples from a Bernoulli distribution $(u^1, \ldots, u^{n_a} \overset{\text{iid}}{\sim} Ber(p))$ for masking probability $p$. Here, $\circ$ denotes element-wise vector multiplication and $\oplus$ denotes vector concatenation.

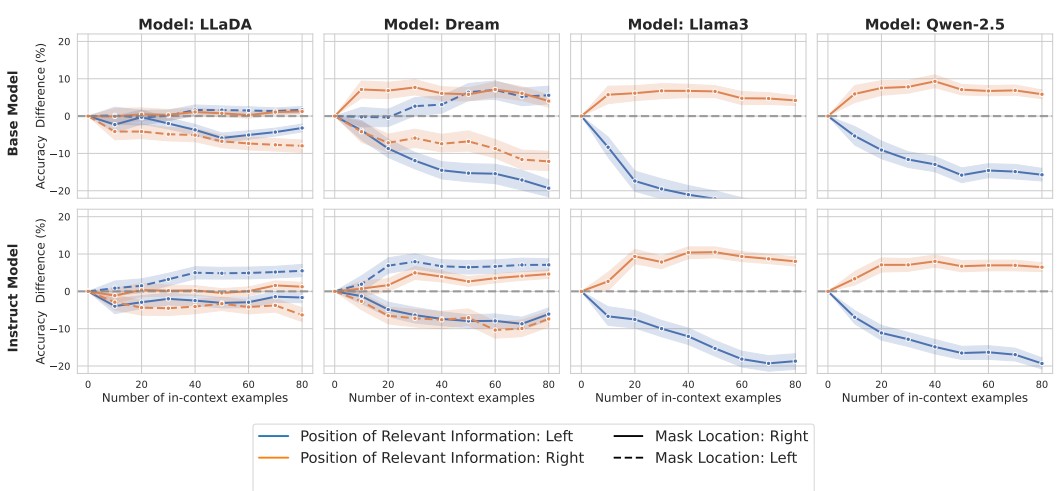

Figure 33: **In the multidimensional classification dataset, across all models, performance degrades when the relevant information is distant from the test question.** We report the accuracy difference when placing relevant information on the left versus randomly (blue line), and on the right versus randomly (orange line). For DLMs, we additionally vary the position of the masked question–placing it at either the left or right end of the in-context examples (solid vs. dashed lines). Across all models performance consistently drops when the relevant information is far from the masked question (blue solid and orange dashed lines), with the effect being most pronounced in ARLMs. Notably, Dream exhibits a stronger recency bias when the masked question is positioned on the right than on the left, suggesting an underlying AR bias. *Shaded regions indicate 95% confidence intervals computed across the 4 dataset types, and 5 seeds.*

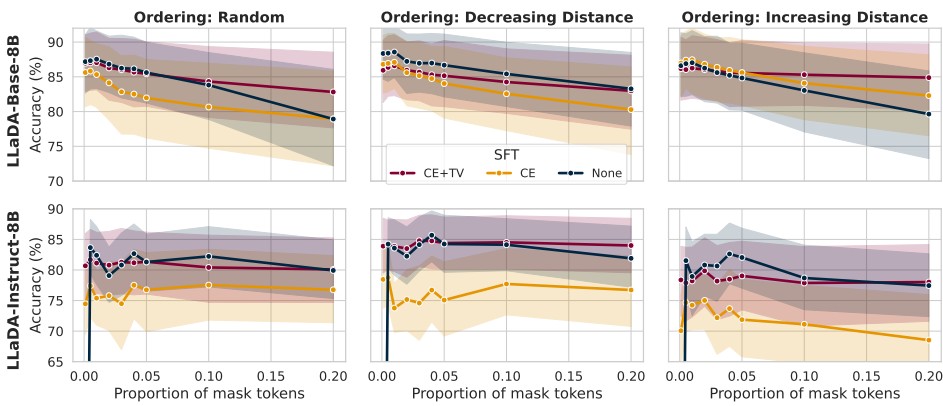

Figure 34: **In the multidimensional-classification dataset, the MA loss prevents performance degradation with the extra masks (particularly for the Base model).** We observe how the performance of the models changes under different ordering schemes of the in-context examples: random ordering, ordering by decreasing distance (most relevant information is located close to the test question) and ordering by increasing distance (most relevant information is located far from the test question). *Shaded regions indicate 95% confidence intervals computed across the 4 dataset types, and 5 seeds.*

Let $\boldsymbol{q} \oplus \tilde{\boldsymbol{a}}$ denote the concatenation of the prompt and the noised answer tokens. To compute our MA loss, we construct two alternative versions of this input, with different numbers of mask tokens appended. That is, we select $l_1, l_2 \in \mathbb{Z}$ randomly without replacement from the range $[0, N - (n_a + n_q)]$, where $N$ is some pre-defined maximum context length. We then construct two inputs: $\boldsymbol{x_1} = \boldsymbol{q} \oplus \tilde{\boldsymbol{a}} \oplus (m) * l_1 = (x_1^1, \ldots, x_1^{n_q+n_a+l_1})$ and $\boldsymbol{x_2} = \boldsymbol{q} \oplus \tilde{\boldsymbol{a}} \oplus (m) * l_2 = (x_2^1, \ldots, x_2^{n_q+n_a+l_2})$. The corresponding labels (not noised) are: $\boldsymbol{x} = \boldsymbol{q} \oplus \boldsymbol{a} = (x^1, \ldots, x^{n_q+n_a})$. Further, let $\mathcal{A}$ denote the set of indices of the elements of $\boldsymbol{x_1}$ and $\boldsymbol{x_2}$ which correspond to the answer-part of the input. With this notation in hand we can define our loss as follows:

$$\mathcal{L}_{CE} = -\frac{1}{2pn_m} \sum_{i=1,2} \sum_{j \in \mathcal{A}} \mathbb{1}\{x_i^j = m\} \log p_\theta(x^j | \boldsymbol{x_i}),$$

$$\mathcal{L}_{TV} = \frac{p}{n_m} \sum_{j \in \mathcal{A}} \mathbb{1}\{x_1^j = m\} TV\left(p_\theta(x^j | \boldsymbol{x_1}), p_\theta(x^j | \boldsymbol{x_2})\right),$$

where $p_\theta$ is the MDLM distribution and $n_m = \sum_{j \in \mathcal{A}} \mathbb{1}\{x_i^j = m\}$ is the number of masked tokens. Our final MA-loss is then constructed as $\mathcal{L}_{MA} = \alpha \mathcal{L}_{CE} + \beta \mathcal{L}_{TV}$ for scaling parameters $\alpha$ and $\beta$.

The first term (**CE loss**) is a cross-entropy loss on the generated answer, ensuring that the model's predictions match the ground-truth answers regardless of how many additional masks are appended. We scale this term by $1/p$, following the standard masked diffusion objective (Sahoo et al., 2024; Nie et al., 2025). The second term (**TV loss**) is a total variational distance that explicitly encourages the probability distributions of the answer tokens to remain consistent across different masking configurations. We scale this term by $p$ to ensure that the distributions are aligned even when there are scarcely any unmasked tokens in the answer. As we explain in **??**, in this case the generations are less constrained by the neighbouring tokens, and thus similarity under different masking conditions is crucial to ensure robustness. We further divide both terms by $n_m$ to ensure that loss is calculated on a per-token basis (to account for the possible large variations in the answer lengths, and hence in the number of masked tokens per input).

### D.2 OTHER DETAILS

Below, we present the pseudo-code for calculating our MA loss, and list the hyperparameters we used during fine-tuning. We use batch-size of three, and we pad the inputs with the end of sequence tokens to ensure equal lengths of the input. Additionally, to make training more stable, we introduce a curriculum for the lengths of the masks added at the end of the inputs, starting from minimal numbers of extra masks, and reaching 600 masks over 5000 gradient descent steps. As in our language modelling setup $p_\theta$ is a categorical distribution, we compute the TV distance $TV\left(p_\theta(x^j | x_1), p_\theta(x^j | x_2)\right)$ as the L1 distance between the probabilities (after softmax) obtained for the two inputs. We conduct the LoRA-based fine-tuning (and the subsequent evaluations of the fine-tuned models) on the non-quantised version of the LLaDA models, to ensure more stable training.

We use the following specific values of the hyperparameters for individual settings, chosen based on the lowest value of the loss functions achieved across the considered settings:

- **Base model, CE loss:** $\beta = 0.0$, $LR = 10^{-6}$
- **Base model, CE + TV loss:** $\beta = 1.0$, $LR = 10^{-5}$
- **Instruct model, CE loss:** $\beta = 0.0$, $LR = 10^{-5}$
- **Instruct model, CE + TV loss:** $\beta = 100.0$, $LR = 5 \times 10^{-7}$

## E EXPERIMENTAL DETAILS

### E.1 MODELS

Throughout our experiments we use the following open-source model families, all accessed via the Huggingface API:

---

**Algorithm 1** Mask-agnostic training

---

**Require:** $\mathcal{P}$: set of input pairs $(q, a)$
**Require:** $p_l, p_u$: lower and upper probabilities of masking
**Require:** $N$: maximum length of text allowed for the model
**Require:** max_masks: Maximal number of masks to be appended to the input
**Require:** $\alpha, \beta$: regularisation coefficients
   **for** $(q, a)$ in $\mathcal{P}$ **do**
        Sample $p \sim U(p_l, p_u)$.
        Create a noised version of the answer $\tilde{a}$ with masking probability $p$.
        Sample $l_1, l_2 \sim \mathcal{U}\left(0, \min(L - \text{len}(p \oplus a), \text{max\_masks})\right)$.
        $x_1 \leftarrow p \oplus \tilde{a} \oplus ([\text{MASK}] * l_1)$
        $x_2 \leftarrow p \oplus \tilde{a} \oplus ([\text{MASK}] * l_2)$
        Pad $x_1, x_2$ with EOS tokens such that they have equal length.
        $o_1 \leftarrow \text{MDLM}(x_1)$
        $o_2 \leftarrow \text{MDLM}(x_2)$
        Compute the MA loss: $\alpha\mathcal{L}_{TV} + \beta\mathcal{L}_{CE}$.
        **Backpropagate**(Loss).
   **end for**

---

- **LLaDA (Nie et al., 2025):** An 8B diffusion language model pre-trained from scratch using the masked diffusion loss (Sahoo et al., 2024).

- **LLaDA-MoE (Zhu et al., 2025):** A 7B mixture of experts diffusion language model pre-trained from scratch using the masked diffusion loss.

- **Dream (HKU NLP Group):** A 7B diffusion language model, whose weights are initialised from those of an autoregressive Qwen-2.5-7B.

- **Qwen-2.5-7B (Yang et al., 2024; Team, 2024):** A fully AR model.

- **Llama3-8B (AI@Meta, 2024):** A fully AR model, with the architecture similar to that of LLaDA (Nie et al., 2025).

For all models, we use greedy decoding strategy (no sampling). We design all of our experiments in a way such that the correct answer consists of only a single token across all the different models and tokenisers. This is to ensure that our experiments can isolate the context-processing abilities of the different models, without being confounded by the effect of tokenisation and/or decoding schemes. This is particularly relevant for DLMs, for which the number of masks added to the prompt can constitute a strong prior about the answer.

### E.2 QUANTISATION

In the experiments which *did not* involve SFT (for which we opted to use the full models), to ensure computational efficiency, we quantised all models to 4-bit precision using the Quanto library. In Figure 35 and Figure 36 we compare the performance of the quantised and non-quantised models on a single task from the pattern recognition suite, verifying that the quantisation has no significant effect on the models' locality bias, nor on the performance degradation under extra masks.

### E.3 DETAILS OF THE FEW-SHOT LEARNING DATASET

Below, we provide further explanations regarding the generation of the few-shot learning tasks used in the main part of the paper. For the relevant (word) tasks, we generate a list of words spanning different categories. We then create the following 8 relevant tasks, by juxtaposing the words from the target category (e.g. adjective) with words from other categories (e.g. verb):

- choose country (out of countries and names),

- choose country (out of countries and names),

- choose capitalised word (out of capitalised and non-capitalised words),

- choose verb (out of verbs, adjectives, prepositions and objects),

Table 2: Fine-tuning hyperparameters

| Category | Parameter | Description | Value |
|---|---|---|---|
| General | Max Context Length | Maximum number of tokens processed in a single forward pass | 1024 |
| | Lower p | Lower threshold for masking probability | 0.2 |
| | Upper p | Upper threshold for probability in sampling | 0.8 |
| Loss | $\alpha$ | Weight coefficient for the CE loss | 0.1 |
| | $\beta$ | Weight coefficient for the TV loss | See D |
| | Max Steps Mask Curriculum | Number of steps for mask curriculum learning | 5000 |
| | Max Masks | Maximum number of mask tokens that can appended to the sequence | 600 |
| Training | Gradient Accumulation Steps | Number of forward passes before parameter update | 43 |
| | Batch size | Size of each batch | 6 |
| | Mixed Precision | Numerical precision format used during training | bf16 |
| | Max Gradient Norm | Maximum L2 norm of gradients for clipping | 1.0 |
| LoRA | Rank | Dimension of low-rank adaptation matrices | 64 |
| | $\alpha_l$ | Scaling factor for LoRA adaptation | 128 |
| | Dropout | Probability of dropping neurons during training | 0.0 |
| | Learning Rate (LR) | Step size for optimizer updates | See D |
| | Weight Decay | L2 regularization coefficient | 0.0 |

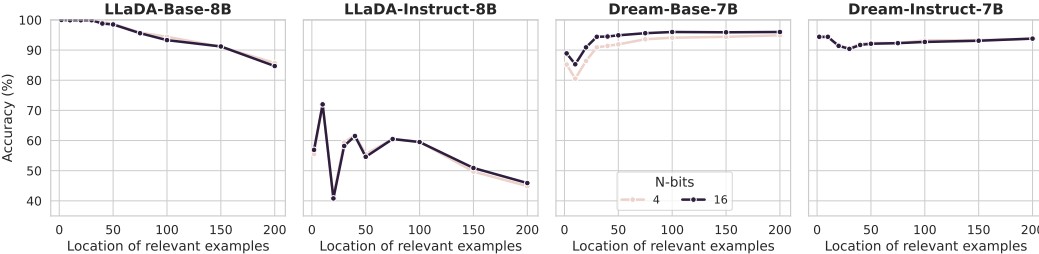

Figure 35: **Quantisation has no significant effect on the performance under varying numbers of mask tokens.**

- choose adjective (out of adjectives, verbs, prepositions and objects),

- choose animal (out of animals, objects, fruits and sports),

- choose colour (out of colours, animals and objects),

- choose emotion (out of emotions, colours, objects and animals),

- choose object (out of objects, emotions, colours and adjectives).

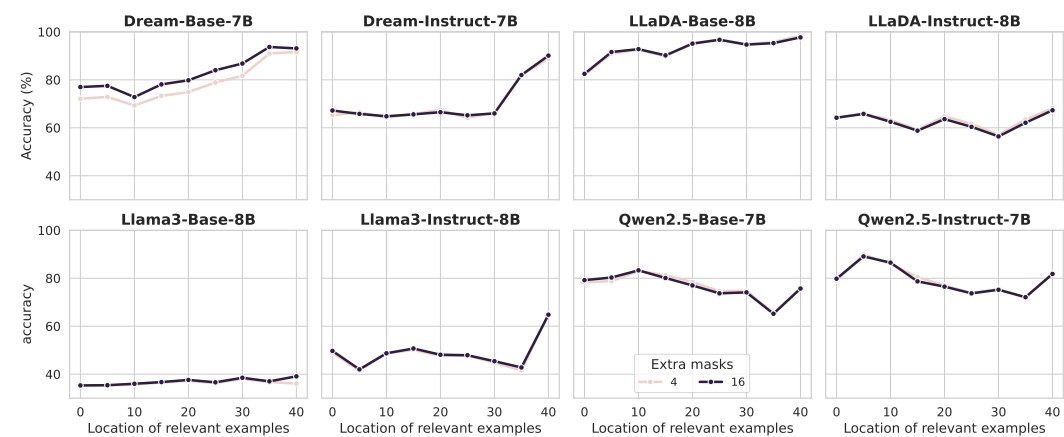

Figure 36: **Quantisation has no significant effect on the the locality of the models.**

Additionally, we consider the following distractor (number tasks), where the candidate numbers are integers sampled without replacement from the range 1 to 1000:

- choose smallest number,
- choose largest number.

Each task contains three possible answers (A, B, C) formatted in a way presented in Section 2. To provide further illustration of the dataset considered, below we include an example of the input obtained for a dataset with task "choose verb" and distractor task "choose smallest number", in the settings when the relevant and distractor tasks are mixed (as in Figure 3) or not (as in Figure 1).

```
'Options: (A) 915, (B) 491, (C) 266\nAnswer:[C].\n\nOptions: (A) 610, (B)
    222, (C) 307\nAnswer:[B].\n\nOptions: (A) 576, (B) 510, (C) 31\
    nAnswer:[C].\n\nOptions: (A) 463, (B) 142, (C) 797\nAnswer:[B].\n\
    nOptions: (A) arrive, (B) thoughtful, (C) near\nAnswer:[A].\n\
    nOptions: (A) 941, (B) 371, (C) 341\nAnswer:[C].\n\nOptions: (A) 694,
     (B) 772, (C) 727\nAnswer:[A].\n\nOptions: (A) tall, (B) compete, (C)
     silly\nAnswer:[B].\n\nOptions: (A) 809, (B) 293, (C) 663\nAnswer:[B
    ].\n\nOptions: (A) 755, (B) 63, (C) 166\nAnswer:[B].\n\nOptions: (A)
    450, (B) 398, (C) 750\nAnswer:[B].\n\nOptions: (A) 541, (B) 698, (C)
    124\nAnswer:[C].\n\nOptions: (A) 289, (B) 567, (C) 774\nAnswer:[A].\n
    \nOptions: (A) reliable, (B) search, (C) zucchini\nAnswer:[B].\n\
    nOptions: (A) 289, (B) 373, (C) 197\nAnswer:[C].\n\nOptions: (A) 402,
     (B) 785, (C) 467\nAnswer:[A].\n\nOptions: (A) 555, (B) 287, (C) 607\
    nAnswer:[B].\n\nOptions: (A) 302, (B) 102, (C) 265\nAnswer:[B].\n\
    nOptions: (A) 790, (B) 409, (C) 904\nAnswer:[B].\n\nOptions: (A)
    deliver, (B) graceful, (C) sensitive\nAnswer:[A].\n\nOptions: (A)
    143, (B) 388, (C) 159\nAnswer:[A].\n\nOptions: (A) 52, (B) 285, (C)
    847\nAnswer:[A].\n\nOptions: (A) 688, (B) 588, (C) 426\nAnswer:[C].\n
    \nOptions: (A) 752, (B) 680, (C) 295\nAnswer:[C].\n\nOptions: (A) 24,
     (B) 868, (C) 400\nAnswer:[A].\n\nOptions: (A) 865, (B) 455, (C) 497\
    nAnswer:[B].\n\nOptions: (A) 214, (B) 506, (C) 469\nAnswer:[A].\n\
    nOptions: (A) 242, (B) 138, (C) 689\nAnswer:[B].\n\nOptions: (A) 159,
     (B) 51, (C) 824\nAnswer:[B].\n\nOptions: (A) 436, (B) 773, (C) 587\
    nAnswer:[A].\n\nOptions: (A) 95, (B) 312, (C) 390\nAnswer:[A].\n\
    nOptions: (A) 30, (B) 982, (C) 727\nAnswer:[A].\n\nOptions: (A) 323,
    (B) 590, (C) 480\nAnswer:[A].\n\nOptions: (A) 640, (B) 621, (C) 525\
    nAnswer:[C].\n\nOptions: (A) 464, (B) 836, (C) 125\nAnswer:[C].\n\
    nOptions: (A) 759, (B) 278, (C) 491\nAnswer:[B].\n\nOptions: (A) 70,
    (B) 435, (C) 386\nAnswer:[A].\n\nOptions: (A) jar, (B) kiss, (C)
    thoughtful\nAnswer:[B].\n\nOptions: (A) 733, (B) 603, (C) 211\nAnswer
    :[C].\n\nOptions: (A) 73, (B) 48, (C) 876\nAnswer:[B].\n\nOptions: (A
    ) passionate, (B) lettuce, (C) master\nAnswer:[C].\n\nOptions: (A)
    169, (B) 784, (C) 919\nAnswer:[A].\n\nOptions: (A) lucky, (B) train,
```

```
    (C) igloo\nAnswer:[B].\n\nOptions: (A) for, (B) calculate, (C) cube\
    nAnswer:[B].\n\nOptions: (A) 861, (B) 579, (C) 735\nAnswer:[B].\n\
    nOptions: (A) 844, (B) 207, (C) 774\nAnswer:[B].\n\nOptions: (A) 502,
     (B) 361, (C) 954\nAnswer:[B].\n\nOptions: (A) innocent, (B) relax, (
    C) upbeat\nAnswer:[B].\n\nOptions: (A) underneath, (B) kill, (C)
    spicy\nAnswer:[B].\n\nOptions: (A) 935, (B) 501, (C) 459\nAnswer:[C
    ].\n\nOptions: (A) concerning, (B) hate, (C) painting\nAnswer:['
```

Listing 1: Example of the input for the few shot learning tasks, with the relevant task "choose verb" and the distractor task "choose smallest number", in the case when the examples are mixed.

```
1 'Options: (A) deliver, (B) graceful, (C) sensitive\nAnswer:[A].\n\
    nOptions: (A) innocent, (B) relax, (C) upbeat\nAnswer:[B].\n\nOptions
    : (A) jar, (B) kiss, (C) thoughtful\nAnswer:[B].\n\nOptions: (A)
    reliable, (B) search, (C) zucchini\nAnswer:[B].\n\nOptions: (A)
    arrive, (B) thoughtful, (C) near\nAnswer:[A].\n\nOptions: (A)
    passionate, (B) lettuce, (C) master\nAnswer:[C].\n\nOptions: (A)
    lucky, (B) train, (C) igloo\nAnswer:[B].\n\nOptions: (A) underneath,
    (B) kill, (C) spicy\nAnswer:[B].\n\nOptions: (A) for, (B) calculate,
    (C) cube\nAnswer:[B].\n\nOptions: (A) tall, (B) compete, (C) silly\
    nAnswer:[B].\n\nOptions: (A) 555, (B) 287, (C) 607\nAnswer:[B].\n\
    nOptions: (A) 463, (B) 142, (C) 797\nAnswer:[B].\n\nOptions: (A) 289,
     (B) 567, (C) 774\nAnswer:[A].\n\nOptions: (A) 464, (B) 836, (C) 125\
    nAnswer:[C].\n\nOptions: (A) 861, (B) 579, (C) 735\nAnswer:[B].\n\
    nOptions: (A) 844, (B) 207, (C) 774\nAnswer:[B].\n\nOptions: (A) 755,
     (B) 63, (C) 166\nAnswer:[B].\n\nOptions: (A) 502, (B) 361, (C) 954\
    nAnswer:[B].\n\nOptions: (A) 52, (B) 285, (C) 847\nAnswer:[A].\n\
    nOptions: (A) 576, (B) 510, (C) 31\nAnswer:[C].\n\nOptions: (A) 242,
    (B) 138, (C) 689\nAnswer:[B].\n\nOptions: (A) 541, (B) 698, (C) 124\
    nAnswer:[C].\n\nOptions: (A) 159, (B) 51, (C) 824\nAnswer:[B].\n\
    nOptions: (A) 610, (B) 222, (C) 307\nAnswer:[B].\n\nOptions: (A) 302,
     (B) 102, (C) 265\nAnswer:[B].\n\nOptions: (A) 915, (B) 491, (C) 266\
    nAnswer:[C].\n\nOptions: (A) 694, (B) 772, (C) 727\nAnswer:[A].\n\
    nOptions: (A) 733, (B) 603, (C) 211\nAnswer:[C].\n\nOptions: (A) 214,
     (B) 506, (C) 469\nAnswer:[A].\n\nOptions: (A) 809, (B) 293, (C) 663\
    nAnswer:[B].\n\nOptions: (A) 865, (B) 455, (C) 497\nAnswer:[B].\n\
    nOptions: (A) 450, (B) 398, (C) 750\nAnswer:[B].\n\nOptions: (A) 323,
     (B) 590, (C) 480\nAnswer:[A].\n\nOptions: (A) 688, (B) 588, (C) 426\
    nAnswer:[C].\n\nOptions: (A) 169, (B) 784, (C) 919\nAnswer:[A].\n\
    nOptions: (A) 790, (B) 409, (C) 904\nAnswer:[B].\n\nOptions: (A) 30,
    (B) 982, (C) 727\nAnswer:[A].\n\nOptions: (A) 73, (B) 48, (C) 876\
    nAnswer:[B].\n\nOptions: (A) 402, (B) 785, (C) 467\nAnswer:[A].\n\
    nOptions: (A) 289, (B) 373, (C) 197\nAnswer:[C].\n\nOptions: (A) 935,
     (B) 501, (C) 459\nAnswer:[C].\n\nOptions: (A) 24, (B) 868, (C) 400\
    nAnswer:[A].\n\nOptions: (A) 436, (B) 773, (C) 587\nAnswer:[A].\n\
    nOptions: (A) 143, (B) 388, (C) 159\nAnswer:[A].\n\nOptions: (A) 640,
     (B) 621, (C) 525\nAnswer:[C].\n\nOptions: (A) 941, (B) 371, (C) 341\
    nAnswer:[C].\n\nOptions: (A) 95, (B) 312, (C) 390\nAnswer:[A].\n\
    nOptions: (A) 70, (B) 435, (C) 386\nAnswer:[A].\n\nOptions: (A) 752,
    (B) 680, (C) 295\nAnswer:[C].\n\nOptions: (A) 759, (B) 278, (C) 491\
    nAnswer:[B].\n\nOptions: (A) concerning, (B) hate, (C) painting\
    nAnswer:['
```

Listing 2: Example of the input for the few shot learning tasks, with the relevant task "choose verb" and the distractor task "choose smallest number", in the case when the examples are not mixed, and the relevant examples are at position 0.0.

### E.4 FORMATTING OF THE IN-CONTEXT LEARNING EXAMPLES

Throughout each experiment, we pre-select a certain group of examples from the specified train set to serve as the in-context learning examples for all the test examples (that is, each test example sees exactly the same in-context learning examples, put in the same order). We always embed the final answer within the square brackets to avoid issues around tokenisation of spaces. For instruct models,

each in-context example is formatted as a pair of messages: a user message containing the question and an assistant message containing the answer. The test question is added as the final user message, with the answer prefix included in the assistant's response.

**Autoregressive models.** For ARLMs, we add the full test question and the beginning of the answer (e.g., `"Label:["`) to the final formatted prompt and ask the model to continue the generation. We always decode only one new token.

**Diffusion models.**

- In section 3: to allow to robustly compare performance between different locations of the masked question within the provided in-context examples, we structure the answer of the masked question as `"Answer:[<|mask|>]."` and add this to the prompt, where `<|mask|>` is the textual representation of the mask token, specific to each MDLM. We add exactly one copy of the mask token in between the square brackets. We also use this setup in the experiments with extra dots, appending the dots *after* the closing the bracket of the answer.

- In section 4 and 5 (as well as in other experiments with varying number of extra masks), we use a generation style more resembling the setup for ARLMs, where we add the full test question and the beginning of the answer (e.g., `"Answer:["`) to the final formatted prompt, followed by the specified number of masks. As the closing bracket `].` is typically tokenised as a single token, using this setup with exactly two extra masks allows us to mostly recover the performance seen in the previous setup.

**Extracting answers.** To evaluate the models' accuracy, we perform string matching on the greedily decoded answer (that is, we perform evaluation on the decoded answer, rather than on the generated tokens).

**Changing the location of the masked question.** In Figure 10, to evaluate the sensitivity of the DLMs to the positioning of the mask, we design experiments in which the masked question is placed at different positions within the in-context examples (at the beginning (left) or end (right)).

