# OpenReview forum: "Masks Can Be Distracting: On Context Comprehension in Diffusion Language Models"
_ICLR.cc/2026/Workshop/Sci4DL — Sci4DL 2026_

### Official Review · Reviewer_B7P1 · 2026-02-26

**Fit:** 3
**Significance:** 3
**Confidence:** 2

**Summary:**

The paper analyzes how context is used in MDLMs during inference. They present two findings, the first that MDLMs have a locality bias, where the authors show that the performance of MDLMs degrade as the relevant information moves further away from the prediction target. Secondly, the number of mask tokens at inference has an important impact on the downstream performance, where increasing the number of mask tokens leads to degradation in performance especially in long horizon tasks.The authors introduce a mask-agnostic fine-tuning loss which when trained with, shows improvement in locality bias and robustness to number of mask tokens.

**Strengths:**

1. The paper conducts careful and systematic ablations to pinpoint issues with the locality bias and high performance variation with changes in mask tokens. They also include two model families: train from scratch (LLaDA) and AR initliazed (Dream) which helps in isolating the effects across model architectures as well.
2. The authors introduce a simple yet novel idea of using a mask agnostic training loss, forcing the invariance of the number of mask tokens at inference time which is an easy way to improve the robustness and also the locality bias of MDLMs.

**Suggestions:**

1. The evaluation is conducted only on multiple-choice questions, where the number of mask tokens and locality bias might have a different behaviour than more free-flowing text completion tasks.
2. To test the hypothesis whether the [MASK] tokens lead to degraded performance just because of repititions of the same token, the authors provide an alternative token by using the '.' token as well. While this is a useful control, using other tokens like frequent words/<pad> tokens etc might help strengthen the arguement further.
2. There are several broken references to figures in the paper, this makes it hard to read, it will be great if the authors can fix that in the manuscript.

---

### Official Review · Reviewer_1byx · 2026-02-27

**Fit:** 3
**Significance:** 2
**Confidence:** 2

**Summary:**

This paper studies how well Masked Diffusion Language Models (MDLMs) comprehend context, and finds two notable limitations. First, despite their global denoising objective, MDLMs still exhibit a strong locality bias, performing best when relevant information sits close to the prediction target. Second, appending extra mask tokens (which is necessary for generation when the answer length is unknown) actively degrades performance, especially in long-context settings. The authors run a clean set of ablations showing that this degradation is specific to mask tokens rather than repeated tokens in general, and that iterative unmasking can recover lost accuracy. To address the issue, they propose a mask-agnostic (MA) fine-tuning loss that combines cross-entropy with a TV-distance term to enforce prediction invariance across different mask counts. This substantially improves robustness on LLaDA models.

**Strengths:**

1. The core question is well-motivated and timely. MDLMs are often motivated by their bidirectional, non-sequential training objective, so it's important to check whether that actually translates into more uniform context usage. The finding that it doesn't is a useful reality check for the community and for understanding how deep learning methods work.
2. The Dream vs. LLaDA comparison is one of the more insightful aspects. Dream's robustness to extra masks, likely stemming from its AR initialization, suggests that the vulnerability is tied to learning from scratch with the masked diffusion loss rather than being an inherent architectural problem.

**Suggestions:**

1. The hypothesis that the locality bias stems from the 1/p weighting in the masked diffusion loss is well-reasoned, and the comparisons between LLaDA and Dream provide indirect support. That said, a more direct test (for instance, training with a modified weighting schedule) would make this claim more definitive.
2. It would be interesting to see whether specific attention heads are responsible for the mask distraction effect. For instance, are there heads that preferentially attend to appended masks instead of context tokens, or does the presence of masks cause a more diffuse collapse in attention sparsity across the board? In combination with the the gradient attribution, the direct attention map visualizations could give a more fine-grained picture of what's going wrong mechanistically.
3. References to many of the figures are broken (i.e. line 94)

---

### Meta-Review · Area_Chair_UgT7 · 2026-03-02

**Recommendation:** Accept

**Metareview:**

This paper empirically studies context comprehension in MDLM, revealing two major limitations: strong locality bias and performance degradation when appending extra mask tokens. To address this issue, the authors introduce a mask-agnostic fine-tuning loss that improves the models' robustness. The reported phenomena and the proposed solution can be of interest for the workshop.

---

### Decision · Program_Chairs · 2026-03-02

Accept